# The Significance of SPP1 in Lung Cancers and Its Impact as a Marker for Protumor Tumor-Associated Macrophages

**DOI:** 10.3390/cancers15082250

**Published:** 2023-04-12

**Authors:** Eri Matsubara, Hiromu Yano, Cheng Pan, Yoshihiro Komohara, Yukio Fujiwara, Shukang Zhao, Yusuke Shinchi, Daisuke Kurotaki, Makoto Suzuki

**Affiliations:** 1Department of Cell Pathology, Graduate School of Medical Sciences, Kumamoto University, Kumamoto 860-8556, Japan; 2Department of Thoracic Surgery and Breast Surgery, Faculty of Life Sciences, Kumamoto University, Kumamoto 860-8556, Japan; 3Center for Metabolic Regulation of Healthy Aging, Kumamoto University, Kumamoto 860-8556, Japan; 4Laboratory of Chromatin Organization in Immune Cell Development, International Research Center for Medical Sciences, Kumamoto University, Kumamoto 860-8556, Japan

**Keywords:** lung cancer, macrophage, osteopontin, SPP1, tumor microenvironment

## Abstract

**Simple Summary:**

Macrophages that infiltrate the cancer microenvironment are referred to as tumor-associated macrophages (TAMs). Phosphoprotein 1 (SPP1), also known as osteopontin, is a multifunctional secreted phosphorylated glycoprotein. In this review, we summarize the significance of TAMs in lung cancers and discuss the importance of SPP1 as a new marker for the protumor subpopulation of TAMs in lung adenocarcinoma. (1) The presence of TAMs is reportedly a poor prognostic factor in lung cancer. (2) We recently revealed that SPP1 expression on TAMs is correlated with poor prognosis and chemoresistance in lung adenocarcinoma. (3) SPP1 is a potentially useful marker for detecting monocyte-derived TAMs.

**Abstract:**

Macrophages are a representative cell type in the tumor microenvironment. Macrophages that infiltrate the cancer microenvironment are referred to as tumor-associated macrophages (TAMs). TAMs exhibit protumor functions related to invasion, metastasis, and immunosuppression, and an increased density of TAMs is associated with a poor clinical course in many cancers. Phosphoprotein 1 (SPP1), also known as osteopontin, is a multifunctional secreted phosphorylated glycoprotein. Although SPP1 is produced in a variety of organs, at the cellular level, it is expressed on only a few cell types, such as osteoblasts, fibroblasts, macrophages, dendritic cells, lymphoid cells, and mononuclear cells. SPP1 is also expressed by cancer cells, and previous studies have demonstrated correlations between levels of circulating SPP1 and/or increased SPP1 expression on tumor cells and poor prognosis in many types of cancer. We recently revealed that SPP1 expression on TAMs is correlated with poor prognosis and chemoresistance in lung adenocarcinoma. In this review, we summarize the significance of TAMs in lung cancers and discuss the importance of SPP1 as a new marker for the protumor subpopulation of monocyte-derived TAMs in lung adenocarcinoma. Several studies have shown that the SPP1/CD44 axis contribute to cancer chemoresistance in solid cancers, so the SPP1/CD44 axis may represent one of the most critical mechanisms for cell-to-cell communication between cancer cells and TAMs.

## 1. Introduction

Lung cancer remains one of the most common causes of cancer deaths worldwide, with an estimated 2 million new cases and 1.76 million deaths per year [1,2]. Cytotoxic anticancer drugs have long played a central role in drug therapies used to treat unresectable advanced lung cancer [1,2]. However, since the 2000s, new drug treatments such as molecular-targeted drugs and immune checkpoint inhibitors have appeared [1,2,3,4,5,6,7]. The efficacy of these drugs in comparison with cytotoxic anticancer drugs has been demonstrated [3,4,5,6,7]. Immune checkpoint inhibitors that have been approved for use in Japan since 2015 include antibodies that target immune checkpoint molecules such as PD-1/PD-L1 and CTLA-4, which are negative regulators of tumor immunity [5,6,7]. However, individual differences in the efficacy of pharmacotherapies and the frequent development of drug resistance have limited the use of current agents.

Tumors are surrounded by a complex array of components that collectively constitute the tumor microenvironment (TME), including extracellular matrix, blood and lymph vessels, fibroblasts, and immune cells [8,9,10,11]. The TME reportedly plays important roles in tumor progression, metastasis, immunosuppression, and drug resistance [8,9,10,11]. Drug resistance is acquired via complex mechanisms in both primary and metastatic sites, and although these mechanisms involve cell intrinsic and extrinsic factors, the latter seem to have been largely overlooked [8]. Many reports have suggested that the TME plays important roles in multiple aspects of cancer progression, particularly drug resistance [8]. New treatment strategies that target the TME are thus desired.

Tumor-infiltrating lymphocytes (TILs), myeloid-derived suppressor cells (MDSCs), and tumor-associated macrophages (TAMs) are well-characterized, representative immune cells that constitute the TME [8,9,10,11]. TILs include cytotoxic T lymphocytes (CTLs) and regulatory T cells [8,9,10,11,12,13]. A high density of CTLs in the TME is considered a good prognostic factor in several cancers, including triple-negative breast cancer, but a lower correlation has been reported in lung cancer [11,12,13]. MDSCs are functionally defined as immunosuppressive, immature myeloid cells [8,9,10,11,14]. As the current understanding of MDSCs is based on observations from mouse studies, exactly which immune cell populations constitute MDSCs in humans remains a matter of contention [14,15,16]. Macrophages that infiltrate the TME are referred to as TAMs [8,9,10,11,15,16]. TAMs exhibit protumor functions related to invasion, metastasis, and immunosuppression, and an increased density of TAMs is associated with poor clinical course in many cancers [8,9,10,11,15,16]. 

Phosphoprotein 1 (SPP1), also known as osteopontin, is a multifunctional secreted phosphorylated glycoprotein [17,18,19]. SPP1 is produced in a variety of organs, such as the kidneys, liver, brain, lungs, and pancreas [17,18,19]. SPP1 is also present in body fluids such as serum, bovine milk, and human urine [18,19,20]. At the cellular level, SPP1 is present on only a few cell types, such as osteoblasts, fibroblasts, macrophages, dendritic cells, lymphoid cells, and mononuclear cells of the immune system [18,19,20]. SPP1 is also expressed by cancer cells, and previous studies have demonstrated correlations between levels of circulating SPP1 and/or increases in SPP1 expression on tumor cells and poor prognosis in many types of cancer [21,22,23]. SPP1 is well known to be involved in cancer cell growth and resistance to chemoradiotherapy through the induction of epithelial–mesenchymal transition (EMT), autophagy, aberrant glucose metabolism, epigenetic alterations, and reduction of drug uptake. These functions are mediated by activation signals from the PI3K/Akt and MAPK pathways induced by the binding of SPP1 to integrin αvβ3 and CD44 [24].

We recently revealed that SPP1 expression on TAMs correlates with poor prognosis and chemoresistance in lung adenocarcinoma [17]. In this review, we summarize the significance of TAMs in lung cancers and discuss the importance of SPP1 as a new marker for the protumor subpopulation of TAMs in lung adenocarcinoma. In addition, SPP1 has been suggested to be useful in distinguishing monocyte-derived TAMs from resident TAMs. The significance of granulocyte macrophage colony-stimulating factor (GM-CSF)-related signals and the SPP1/CD44 axis in chemoresistance is also discussed.

## 2. The M1/M2 Concept of TAMs in Lung Cancers

Tumor tissue is composed of variable numbers of cancer cells and stromal cells, and macrophages are one of the most abundant types of cancer stromal cells [15,16]. TAMs derived from circulating monocytes and resident macrophages differentiate into several different phenotypes due to the cytokine balance and other factors in the TME [15,16]. TAMs are composed of heterogenous subpopulations. Many researchers, including our group, previously explained the heterogeneity of TAMs using the M1/M2 concept; however, recent studies have suggested that the M1/M2 concept is inadequate for describing human cells, as it is based on studies using mouse models [25].

A number of reports have described the relationships between TAMs and clinical prognosis in lung cancer [26,27,28,29,30,31,32,33,34,35,36,37]. Although a preponderance of these reports indicate that TAMs are correlated with poor prognosis in lung cancer, conflicting results have been reported regarding the prognostic significance of TAMs in tumor tissues [11,16,26]. Among the reasons for these discrepancies, in almost all studies using immunohistochemistry (IHC), lung adenocarcinoma and squamous cell carcinoma (SCC) were evaluated interchangeably [11]. Adenocarcinoma also comprises many histological patterns, such as papillary/acinar type, solid type, and occasionally a complex glandular pattern. Furthermore, different macrophage markers, such as CD68, CD204, and CD163, have been used in some studies [11,26], and TAMs were evaluated without considering their localization. The designation of TAM usually refers to macrophages that infiltrate tumor nests or tumor stroma [15,16]. In the lung, there are exceptional components, such as alveolar macrophages, in the alveolar space of peritumoral areas. These components of TAMs, the cells infiltrating tumor nests and tumor stroma and those in the alveolar spaces, may need to be evaluated separately, as they exhibit many differences in distribution, morphology, and gene expression [27].

## 3. Function and Significance of TAMs in Lung Cancers

Several reports have described the relationship between TAMs and prognosis, but they are limited to adenocarcinoma [26,27,28,29,30]. Ohtaki et al. reported that the presence of CD204-positive TAMs in cancer stroma is related to poor prognosis. They reported that levels of monocyte chemoattractant protein-1 and IL-10 are significantly correlated with the number of CD204-positive TAMs, which can induce the differentiation, accumulation, and migration of M2 TAMs [28]. An increased density of TAMs has also been demonstrated in males, smokers, and patients with tumors showing histologically solid type, higher stage, or lymphovascular invasion [28]. Kaseda et al. reported that the presence of histological vascular invasion with abundant CD204-positive TAMs is predictive of a high likelihood of recurrence [29]. Detection of fibroblasts and endothelial cells has also been shown to be associated with recurrence, and this study was the first report to describe the TME of intravascular cancer. Ito et al. demonstrated that a high density of podoplanin-positive fibroblasts and CD204-positive TAMs predicted worse clinical course in Stage I lung adenocarcinoma, suggesting the significant involvement of cell-to-cell communications by fibroblasts and TAMs in cancer invasion [30]. Although non-detailed histological subtype was described, an increased density of CD163-positive TAMs either in cancer nests or stroma correlated with poor clinical course [31]. Studies in Stage III/IV patients treated with epidermal growth factor receptor (EGFR)-tyrosine kinase inhibitors (TKIs) have shown high numbers of CD163-positive TAMs were associated with poor clinical course, and TAMs were suggested to be involved in the resistance to treatment using EGFR-TKIs [32]. However, other reports have indicated that TAMs are not correlated with prognosis [33,34]. Carus et al. reported that there is no correlation between prognosis in lung adenocarcinoma and the degree of infiltration of tumor nests and the presence of stromal CD163-positive macrophages [33]. As CD163 can also be detected in lung cancer cells by immunohistochemistry, analysis of CD163 might not be adequate to evaluate TAMs in lung cancers [38]. Although detailed mechanisms for this phenomenon that CD163 stained positively in cancer cells have been uncovered, soluble CD163 was suggested to be accumulated in cancer cells via unknown receptors. Descriptions of CD163-positive cancer cells have been reported for several solid cancers, including breast cancer and colorectal cancer [39,40].

There are fewer reports about the relationship between TAMs and prognosis in SCC and neuroendocrine carcinoma relative to the number of reports regarding adenocarcinoma. Hirayama et al. demonstrated that the number of CD204-positive TAMs in the tumor stroma, but not in the tumor nest, is correlated with poor prognosis in SCC, especially in pStage I [35]. Furthermore, they reported that the number of CD204-positive TAMs is strongly correlated with microvessel density, which might be related to monocyte chemotactic protein 1 (MCP-1) activity [35]. With regard to neuroendocrine carcinomas, Takahashi et al. reported that the number of infiltrating CD204-positive macrophages does not affect outcome in patients with high-grade neuroendocrine carcinomas (large cell neuroendocrine carcinoma and small cell lung carcinoma) [36]. However, another study suggested that cell-to-cell interactions between neuroendocrine carcinoma cells and TAMs contribute to disease progression and chemoresistance via STAT3 activation [37].

## 4. Significance of SPP1 in Lung Cancer

A number of previous studies have demonstrated correlations between levels of circulating SPP1 and/or increased SPP1 expression on tumor cells and poor prognosis in various cancers, including non-small cell lung cancer (NSCLC) ([41,42,43,44,45,46,47], summarized in Appendix A). In general, analyses to detect SPP1 can be categorized into two primary subgroups: enzyme-linked immunosorbent assay (i.e., ELISA) of blood samples and IHC analysis of cancer tissue samples.

Several reports have discussed SPP1 in serum or plasma of lung cancer patients [44,48,49,50,51,52,53]. Most of these reports demonstrated that circulating SPP1 is correlated with poor prognosis in NSCLC patients [44,48,49,50,51,52,53]. Mack et al. reported that low plasma levels of SPP1 are significantly associated with improved clinical outcomes in chemotherapy-treated patients with advanced NSCLC [48]. Interestingly, they found that tumor SPP1 expression, as measured by IHC, is not correlated with plasma SPP1 level and patient outcome, and they hypothesized that non-malignant cells may contribute to plasma SPP1 concentrations [48]. With regard to circulating SPP1, more data are available for pleural mesothelioma than for lung cancer [54,55]. Pass et al. reported that serum SPP1 levels can be used to distinguish individuals with exposure to asbestos who do not have cancer from those with exposure to asbestos who have pleural mesothelioma [54].

Several reports have discussed SPP1 expression on cancer cells in lung cancer patients [41,42,43,44,45,46,47]. As with circulating SPP1, high expression of SPP1 on cancer cells is correlated with poor prognosis [41,42,43,44,45,46,47]. However, in almost all reports using IHC, lung adenocarcinoma and SCC were evaluated interchangeably. Furthermore, the SPP1 staining evaluation method also varied depending on the study. Several reports have also examined the association between SPP1 and poor prognosis [56,57,58,59]. Wang et al. reported that the promotion of stem-like features mediated by the SPP1-EGFR pathway is associated with the development of radiation resistance in KRAS-mutated lung cancer [56].

## 5. Significance of SPP1 in TAMs

Most reports that discuss SPP1 expression in lung cancer have focused specifically on expression in cancer cells. However, as mentioned earlier, SPP1 is also expressed on other cell types, such as osteoblasts, fibroblasts, macrophages, dendritic cells, lymphoid cells, and mononuclear cells of the immune system [19,20,21]. Although the proportion and importance of each type of cell that infiltrates cancer tissues vary depending on the case and tissue type, macrophages are the predominant cells and play an important role in cancer tissues [15,16]. The expression of SPP1 by macrophages has attracted considerable attention recently [60,61,62,63,64,65,66,67]. We evaluated the expression of SPP1 by TAMs in lung cancer specimens (228 cases of adenocarcinoma and 103 cases of squamous cell carcinoma) using double-IHC with anti-SPP1 and anti-macrophage antibodies [17]. In IHC analysis, SPP1 staining was stronger in TAMs than in cancer cells (Figure 1A). Interestingly, high expression of SPP1 in TAMs was associated with worse clinical course in patients with adenocarcinoma, especially in cases without EGFR mutations. In contrast, there was no significant association between SPP1 expression in cancer cells and clinical course. TAMs in adenocarcinoma were morphologically divided into two populations: macrophages with foamy-like and large-scale cytoplasm located in cancer nests and glandular structures versus macrophages with a smaller-sized and spindle-shaped cytoplasm located in stromal areas (Figure 1B). Macrophages detected in cancer nests and stroma were referred to as TAMs, and macrophages in peritumoral areas were referred to as alveolar macrophages. A significant difference was observed in terms of the size of macrophages among the three groups (Figure 1B). We also observed high expression of SPP1 on intratumoral macrophages but low expression on peritumoral macrophages, with no expression at all on macrophages distant from tumors (Figure 1D). Consistent with these observations, it was recently reported that TAM-SPP1 is highly expressed in intratumoral TAMs but absent in TAMs in the invasive front and alveolar macrophages in non-cancerous areas [68].

## 6. Significance of SPP1 in TAMs: A New Perspective from Single-Cell Analysis

A recent human lung cancer study using single-cell RNA-seq revealed that myeloid cells in the TME can be divided into four groups [68]. Group I cells include tissue-resident macrophages that express high levels of cell cycle-related genes; group III cells include CD14-positive monocytes; group IV cells include CD16-positive monocytes; and the remaining cells are broadly classified into group II. These group II cells are thought to be derived from circulating monocytes, as they express mature macrophage genes and lack the transcripts expressed by group I cells (Figure 2A,B). Additionally, group I macrophages localize close to tumor cells in the early stage of tumor formation but redistribute to the periphery of the tumor during tumor growth, and the intratumoral area becomes dominated by group II macrophages [68]. Using open access data, we identified several representative genes and created a heat map, which revealed that SPP1 is strongly expressed by group II macrophages [17,68]. These findings suggest that SPP1 is a useful marker for classifying TAMs as either monocyte-derived cells or resident cells (Figure 1C). These data also suggest that monocyte-derived TAMs exhibit stronger protumor effects than resident TAMs and that smaller-sized TAMs detected in stromal areas are derived from tissue-resident macrophages. Multiplex immunofluorescence IHC showed strong expression of SPP1 in areas with a high density of immune cells (Figure 3). SPP1 was highly expressed in CD163-positive TAMs and also CTLs, suggesting that SPP1 is highly expressed in immunologically “hot” areas.

## 7. Possible Involvement of the GM-CSF-SPP1 Loop in Chemoresistance

Our previous study found that GM-CSF strongly induces SPP1 expression in human macrophages [17]. GM-CSF stimulation activates the STAT3/5 pathways; however, which signaling pathway contributes to SPP1 expression in macrophages remains unclear [69,70]. Receptors for GM-CSF are specifically expressed on myeloid-lineage cells but are rarely expressed on cancer cells. However, GM-CSF expression was observed in several cancers, including lung cancer and pancreatic cancer [70,71]. GM-CSF expression in lung adenocarcinoma cells was increased by stimulation with paclitaxel, docetaxel, carboplatin, and pemetrexed, and GM-CSF enhanced PD-L1 overexpression in macrophages via STAT3 signaling [70]. A high density of PD-L1-positive TAMs correlated with a worse clinical course in lung adenocarcinoma cases without EGFR mutation, and anti-GM-CSF antibody abrogated the PD-L1 overexpression in macrophages stimulated with cancer cell supernatant [70]. STAT3 signal was a critical regulator for the GM-CSF/PD-L1 axis in human macrophages [70]. An in vivo study using anti-GM-CSF antibody showed that the neutralization of GM-CSF suppressed LLC tumor growth, and detailed examination indicated that GM-CSF blocking inhibited TAM infiltration and increased CTL infiltration in tumor tissues [70]. However, discrepancies have been identified in the mechanisms related to PD-L1 overexpression in macrophages between humans and mice [70]. A similar study using anti-GM-CSF antibody was reported for pancreatic cancer [71]. GM-CSF secretion was enhanced by gemcitabine treatment, which promoted a GM-CSF-induced change of myeloid cells to an immunosuppressive phenotype [71]. On the basis of these findings, we hypothesized that chemotherapy enhances cancer-cell-derived GM-CSF expression, which activates TAM-derived SPP1 and, in turn, induces chemoresistance in cancer cells (Figure 4).

SPP1 has been shown to be involved in resistance to chemoradiotherapy for several solid cancers, including lung cancer. As mentioned above in the Introduction, SPP1 is known to be associated with EMT, autophagy, aberrant glucose metabolism, epigenetic alteration, and reduction of drug uptake, mediated by activation signals from the PI3K/Akt and MAPK pathways [24]. Increased SPP1 expression and decreased E-cadherin expression were observed in the EMT area of lung adenocarcinoma, and an in vitro study revealed that SPP1-induced EMT was suppressed by inhibitors for PI3K/Akt and MAPK signals [72]. BMI1 is known as a critical molecule related to EMT and cancer stemness, and we identified BMI1 as one of the molecules inducing chemoresistance in lung adenocarcinoma cells [17]. Subsequent studies were unrelated to lung cancer, but many indicated the SPP1-related chemoresistance and radioresistance of cancer cells. SPP1-induced autophagy played a critical role for stem-like properties and chemo-resistance of cancer cells by activating the PI3K/mTOR pathway [73]. SPP1 facilitated cancer cell chemoresistance via the activation of the CD44 receptor, PI3K/AKT signaling, and ATP-binding cassette (ABC) drug efflux transporter activity, and anti-SPP1 and anti-CD44 antibodies improved the sensitivity of cancer cells to cisplatin in a mouse model [74]. Nakamura et al. found SPP1 expression was increased in oral cancer tissues with resistance to combined therapy using 5-FU and radiation [75]. They also showed that blocking of integrin αvβ3 by antibody significantly increased the sensitivity of cancer cells to chemo- and radiotherapy by means of in vitro cell culture study. In lung cancer, increased serum SPP1 concentrations after radiotherapy predicted a worse clinical course, suggesting the possible involvement of SPP1 in radioresistance [76]. Thus, SPP1 is thought to be a promising target for anti-cancer therapy. However, no anti-SPP1 antibodies are clinically available because of the multiple receptor-binding domains of SPP1 [77].

## 8. Potential Immunotherapy against GM-CSF

GM-CSF was originally found as a colony-stimulating factor for granulocytes and macrophages, being later revealed to act on the maturation and differentiation of myeloid cells [78]. GM-CSF-deficient mice developed pulmonary alveolar proteinosis (PAP) as a major phenotype, although steady-state myelopoiesis changed minimally. GM-CSF is reportedly required for the development of alveolar macrophages via the induction of peroxisome-proliferator-activated receptor (PPAR)-γ expression, while macrophage differentiation in other organs was independent of PPAR-γ expression [79]. PPAR-γ expression induced by GM-CSF also upregulated ABCG1, which accelerates cholesterol clearance, and a significant elevation of intracellular cholesterol was observed in alveolar macrophages from PAP patients [80]. High levels of autoantibody are well known to neutralize GM-CSF, causing autoimmune PAP. However, inhibition of GM-CSF-GM-CSFR signaling was suggested to be useful for the treatment of several inflammatory or autoimmune diseases, and neutralizing antibodies for GM-CSF and antibodies blocking GM-CSFR have been under clinical trial [78]. A phase II clinical trial in patients with rheumatoid arthritis showed beneficial effects of anti-GM-CSF antibody on systemic inflammation [81], but no further findings from phase III trials have been reported.

A bidirectional function of GM-CSF in anti-cancer therapy has been suggested. As a critical factor in dendritic cell differentiation, GM-CSF has been used as an immunostimulatory adjuvant to elicit antitumor immunity [82]. A GM-CSF-expressing allogenic cancer cell line has been used as a cancer vaccine for anti-cancer immunotherapy in several clinical trials, but few beneficial effects have been published [83]. GM-CSF was thus expected to enhance the anti-cancer immune system but was also suggested to be involved in immune suppression via inducing immunosuppressive myeloid cells such as myeloid-derived suppressor cells and TAMs [84]. In our previous study using a murine tumor model, anti-GM-CSF antibody significantly suppressed subcutaneous tumor growth of Lewis lung carcinoma (LLC), and a decreased density of TAMs and increased density of CTLs were observed in mice with anti-GM-CSF therapy [70]. Anti-GM-CSF therapy also inhibited the maturation and differentiation of TAMs. However, anti-GM-CSF therapy was ineffective against an E0771 breast cancer model [85]. This discrepancy might have been due to GM-CSF expression in tumor cells, since LLC cells express high levels of GM-CSF, whereas E0771 expresses no GM-CSF. Anti-GM-CSF therapy might be useful for anti-cancer immunotherapies against GM-CSF-expressing cancers. 

## 9. Significance of TAM-Derived SPP1 in Immune Suppression

Myeloid cells, including monocyte/macrophages, myeloid-derived suppressor cells (MDSCs), and neutrophils, are involved in immune suppression via the secretion of many kinds of immunosuppressive factors. These myeloid cells are known to express PD-L1, PD-L2, IL-10, indoleamine 2,3-dioxygenase, prostaglandin E2, and IL-6 [25]. Klement et al. showed that SPP1 derived from myeloid cells suppressed CTL function, and SPP1 expression in myeloid cells was regulated by IRF8 in an animal study [86]. They also demonstrated that IRF8 expression was decreased and SPP1 expression was increased in colon cancer, as well as clarifying the immunosuppressive function of SPP1 in human cells. Sangaletti et al. reported that cancer metastasis was significantly suppressed in SPP1-deficient mice, and cancer-derived SPP1 was related to the immunosuppressive function of MDSCs via regulating immune-related molecules such as arginase 1, nitric oxide synthase 2 (NOS2), vascular endothelial growth factor (VEGF), and IL-6 [87]. Thus, SPP1-related signals are considered to represent a potential target for anti-cancer immunotherapies. PD-L1 expression in macrophages was enhanced by SPP1 stimulation [62]. A significant correlation has been identified between SPP1-expressing TAMs and protumor activation of cancer-associated fibroblasts (CAFs) in colorectal cancer, suggesting SPP1 as a factor in cell-to-cell communications between TAMs and CAFs [88]. In lung cancer, CAFs are considered critical stromal cells supporting cancer progression, and close distributions of TAMs and CAFs have been indicated in several pathological examinations [89]. Cell-to-cell interactions between TAMs and CAFs are thought to play important roles in solid cancer, particularly in the gastrointestinal and respiratory organs [90]. SPP1 is thus potentially involved in immunosuppression in the TME via direct suppressive effects on CTLs and indirect stimulation of the immunosuppressive functions of MDSCs, TAMs, and CAFs.

## 10. Significance of SPP1 Expression in TAMs in Malignant Tumors Other Than Lung Cancer

SPP1 expression in TAMs has also been of interest regarding malignant tumors other than lung cancer. In breast cancer, SPP1 derived from TAMs has been suggested to be involved in cancer cell growth and progression, and single-cell RNA sequence analysis has indicated that SPP1 is highly expressed in monocyte-derived TAMs as compared with resident macrophages/TAMs [91]. Single-cell analysis in breast cancer has also shown that SPP1-positive TAMs express high levels of apolipoprotein E (APOE), CD204, CD68, and CADM1. High densities of SPP1-expressing TAMs and CAF appear associated with resistance to immunotherapy in hepatocellular carcinoma, and animal studies have revealed that SPP1 secreted from TAMs is involved in the infiltration of CAFs and suppression of CTL infiltration into tumor tissues [92]. Cancer stem cells of gastric cancer expressed high levels of CD44 [93]. Marked infiltration of SPP1-expressing TAMs correlated with a worse clinical course in gastric cancer, the SPP1/CD44 axis potentially associated with stemness properties, and intratumoral heterogeneity linked to resistance to anticancer therapies [93]. A similar significance of SPP1/CD44-related signals has also been suggested in glioma and hepatocellular carcinoma [94,95]. In colorectal cancer, SPP1-expressing TAMs were located in the invasive front, and the SPP1 signal was suggested to be involved with cancer cell expression of HLA-G, which in turn was associated with cancer cell escape from the immune system [66]. HLA-G was first found to be involved in immune tolerance by suppressing maternal immune responses to fetal tissues, and recent studies have suggested that HLA-G might represent a promising target for anti-cancer immunotherapies, since increased expression of HLA-G has been reported in several kinds of cancer [96]. SPP1-expressing TAMs also secrete CXCL9, which potentially enhanced infiltration of CTLs and regulatory T cells via CXCR3.

CD44v6, a functional variant of CD44, was overexpressed on cancer cells located in the invasive front of colorectal cancer, and SPP1-expressing TAMs interacted with CD44v6-postive cancer cells, indicating the significance of SPP1/CD44v6 binding in cancer progression [97]. Among CD44 variants, CD44v9 is considered to be associated with cancer stem cells in various cancers, but few reports have described the interactions between CD44v9 and SPP1. In metastatic colorectal cancer, cell-to-cell communication between SPP1-expressing TAMs and CAFs has been suggested to be involved in CTL dysfunction, and SPP1-expressing TAMs are characterized by a foamy cell gene signature [67]. Foamy macrophages are also referred to as lipid-associated macrophages, and lipid-associated TAMs in breast cancer express an M2-like gene signature such as CD163 via the secretion of various protumor molecules [98].

SLC2A1, a hypoxia-related molecule, has been identified as a critical regulator of the immunosuppressive microenvironment in metastatic liver cancer via modulation of the chemotaxis of SPP1-expressing TAMs [99]. SPP1-expressing TAMs are suggested to influence the endothelial–mesenchymal transition, generating a subpopulation of protumor CAFs, and this cell-to-cell communication might be mediated by the SPP1/CD44 axis [100]. In pancreatic cancer, SPP1 was expressed in both cancer cells and TAMs, and hypoxia has been suggested to induce SPP1 expression [101]. 

Several studies using a novel, single-cell RNA sequence have indicated the significance of SPP1 secreted from TAMs, playing important roles in immunosuppression via collaboration with immunosuppressive CAFs. 

## 11. Limitations

In this review, we mainly discussed the function and significance of SPP1 derived from TAMs in lung cancer, but differences might exist between each organ and histological subtype. In addition to this discrepancy in organs, a discrepancy between humans and mice is also indicated.

## 12. Conclusions

In this review, we discussed the significance of TAM-derived SPP1 in the progression of lung adenocarcinoma. SPP1 is not only a prognostic marker for lung adenocarcinoma, but also a marker for monocyte-derived macrophages in lung adenocarcinoma. As SPP1 expression in TAMs is induced by GM-CSF, and GM-CSF is produced by cancer cells stimulated with anti-cancer agents, cell-to-cell communication via GM-CSF and SPP1 is thought to be involved in the development of chemoresistance. SPP1 is a critical factor in the development of a protumor TME. Thus, targeting SPP1-related signaling pathways such as the SPP1/CD44 axis is a promising approach for the treatment of lung adenocarcinoma. 

## Figures and Tables

**Figure 1 cancers-15-02250-f001:**
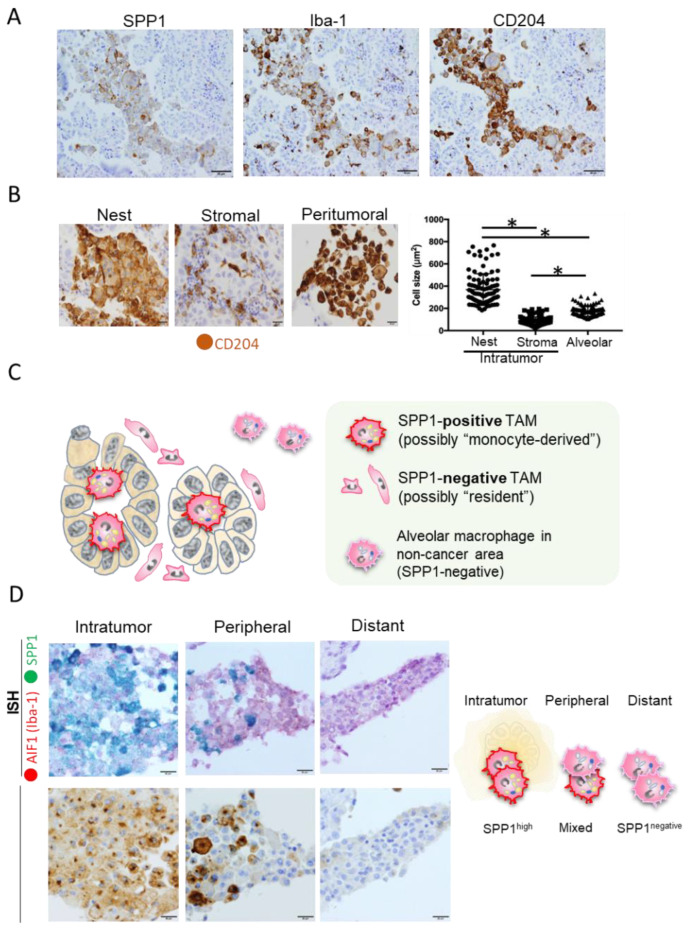
SPP1 expression in macrophages. (**A**) Representative images of IHC staining of SPP1, Iba-1, and CD204 expressions in serial sections of lung adenocarcinoma. Iba-1 and CD204 were used as specific markers for macrophages. Scale bar; 50 μm. (**B**) The size of CD204-positive macrophages infiltrating the cancer nest, stroma, and peritumoral area was evaluated using ImageJ software. Macrophages detected in the cancer nest and stroma were referred to as TAMs, and macrophages in the peritumoral area were referred to as alveolar macrophages. *; *p* value < 0.05. Scale bar; 20 μm. (**C**) Schematic illustration of macrophage phenotypes in and around cancer tumors. (**D**) In situ hybridization (ISH) analyses of AIF1 (red) and SPP1 (green) and IHC of SPP1 were performed in serial section, and figures in intratumor, peripheral, and distant areas are presented. Scale bar; 20 μm.

**Figure 2 cancers-15-02250-f002:**
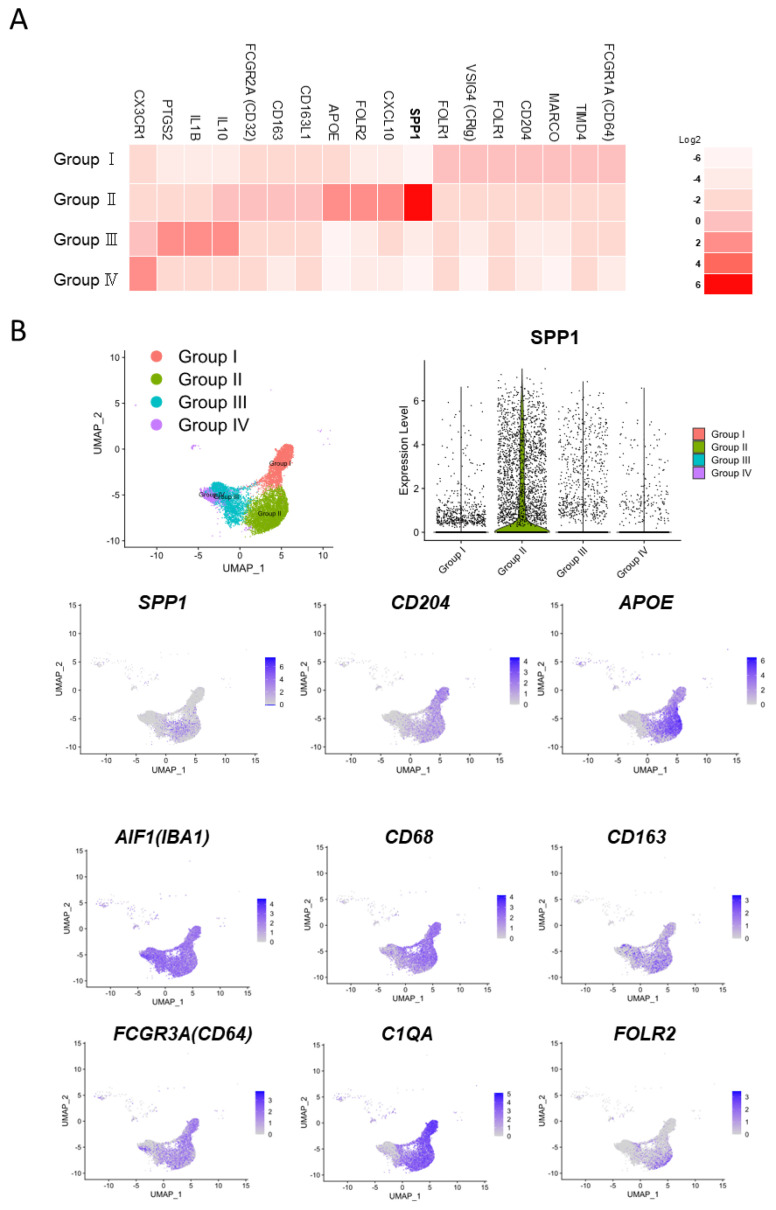
Analysis of single-cell RNA-sequence data of NSCLC. (**A**) Re-analysis of published single-cell RNA sequence data for NSCLC (BioProject accession; PRJNA609924) [68]. Heatmap of the expressions of selected genes in macrophages and monocytes of Groups I, II, III, and IV (tissue-resident macrophages, monocyte-derived macrophages, CD14+ monocytes, and CD16+ monocytes) based on data from an original report examining 30 NSCLC cases. (**B**) UMAP plots of macrophage and monocyte populations (Groups I, II, III, and IV) based on 9 selected cases of adenocarcinoma among the 30 abovementioned NSCLC cases. Expression levels of SPP1 across the four groups are summarized using a violin plot. Expression levels of SPP1, CD204 (MSR1), AIF1 (IBA1), CD68, CD163, FCGR3A, C1QA, FOLR2, and APOE across the four populations, with intensity represented in purple in UMAP plots. Cells are grouped as per the method of the original report.

**Figure 3 cancers-15-02250-f003:**
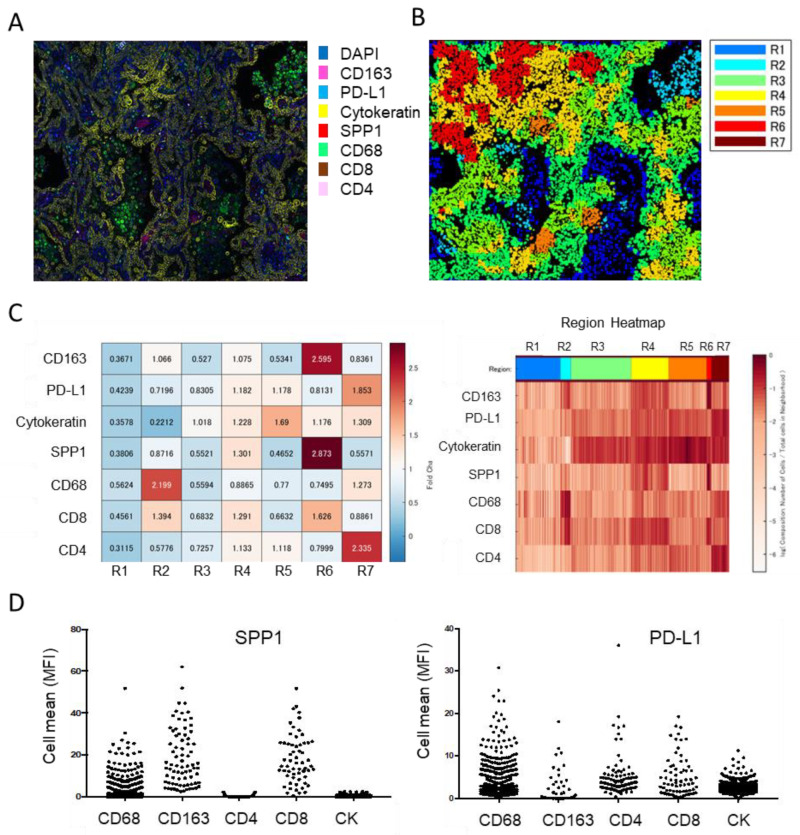
Multiplex immunofluorescent IHC analysis of lung adenocarcinoma. (**A**) IHC of CD163, PD-L1, cytokeratin, SPP1, CD68, CD8, and CD4 performed using the PhenoCycler™ (CODEX) system (AKOYA Bioscience, Marlborough, MA). (**B**) Spatial analysis of cluster neighborhood regions, with regions classified into 7 areas. (**C**) Tanle and Region Heatmap based on mean fluorescence intensity (MFI). MFI is normalized to the maximum MFI per neighborhood. (**D**) SPP1 and PD-L1 expression in each cell type presented in dot-blot graphs.

**Figure 4 cancers-15-02250-f004:**
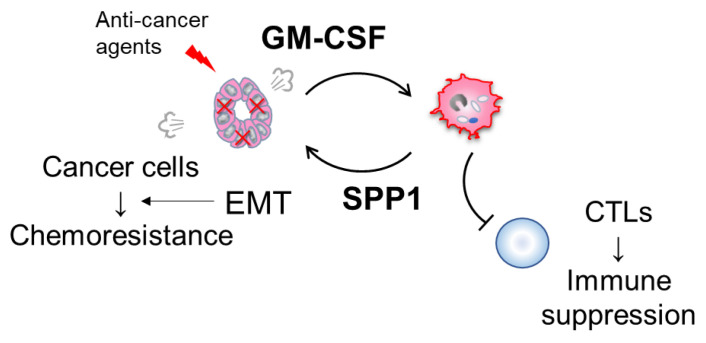
Schematic illustration of possible involvement of the GM-CSF-SPP1 loop in chemoresistance. Anti-cancer agents potentially upregulate GM-CSF secretion from cancer cells. GM-CSF promotes the infiltration of TAMs and a change in TAMs to a protumor phenotype, which in turn induces cancer cell growth and chemoresistance.

## Data Availability

The data presented in this study are available in this article.

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
