# Peer review of "The Significance of SPP1 in Lung Cancers and Its Impact as a Marker for Protumor Tumor-Associated Macrophages"

_cancers, 2023, doi:10.3390/cancers15082250_

Round 1
Reviewer 1 Report
Matsubara and coworkers have revised the recent literature on SSP1( osteopntin) in lung cancer. The review is well organised , the results of the studies are well presented and overall the review is very clear.
I think that the review could benefit of a more detailed discussion of SSP1 as a prognostic marker and as therapeutic target
Author Response
Matsubara and coworkers have revised the recent literature on SSP1( osteopntin) in lung cancer. The review is well organised , the results of the studies are well presented and overall the review is very clear.
I think that the review could benefit of a more detailed discussion of SSP1 as a prognostic marker and as therapeutic target
Response>> We sincerely appreciate to reviewers for kind and critical comments. We added the point in the revised version.
Reviewer 2 Report
This review discusses the function and significance of SPP1 in tumor-associated macrophages of human lung cancers. The review in general is clearly written and complements other recent reviews of SPP1 function in other cancer types. The major concern is that some of the figures appear to present new data or at least new analyses of data previously reported by the authors or others. Therefore, it is unclear whether this is purely a review or whether the new data needs to be supported by including relevant experimental and statistical analysis methods.
Line 119: please clarify “histological slid type”
Line 209 mac-rophages
Figure 1D: Red and green ISH staining is not visible in the composite images. Please add separate red and green channels.
Figure 1B: Is the quantitative data in the graph from a prior publication, reanalysis of published data, or is this new data?
Figure 2 legend: please add a citation to the source publication. Is this data directly from Casanova-Acebes 2021, or did the authors perform new analyses of the published scRNAseq data?
Figure 3: Is the quantitative data in panels C and D derived from the single tissue section shown in A and B? Is this previously published or new data?
Figure 4: What is known regarding mechanisms by which exogenous SPP1 induces chemoresistance in cancer cells? Is a specific receptor involved? Based on the cited antibody blocking, is integrin αvβ3 the relevant receptor?
LLC is used without definition. Lewis lung carcinoma?
Line 360: is a word missing in “In breast cancer, SPP1 derived from TAMs was suggested to cancer cell growth and progression”
Line 364 please correct “High density of SPP1-expres-364 sin TAMs and CAF”
Need to correct several grammar/usage errors including in line 399
The data availability statement may need to be modified unless proper citations are provided for what appear to be primary data presented in the figures. At least need to provide GEO accession numbers for the relevant RNAseq data.
Author Response
This review discusses the function and significance of SPP1 in tumor-associated macrophages of human lung cancers. The review in general is clearly written and complements other recent reviews of SPP1 function in other cancer types. The major concern is that some of the figures appear to present new data or at least new analyses of data previously reported by the authors or others. Therefore, it is unclear whether this is purely a review or whether the new data needs to be supported by including relevant experimental and statistical analysis methods.
Line 119: please clarify “histological slid type”
Response >> We sincerely appreciate to reviewers for kind and critical comments.
Following sentence was added in the text.
“Adenocarcinoma also comprises many histological patterns, such as papillary/acinar type, solid type, and occasionally a complex glandular pattern.”
Line 209 mac-rophages
Response >> This point was corrected in the revised version.
Figure 1D: Red and green ISH staining is not visible in the composite images. Please add separate red and green channels.
Response >> Thanks for the comments. This image is from the double IHC of traditional methods, and therefore, the color signals could not be separated.
Figure 1B: Is the quantitative data in the graph from a prior publication, reanalysis of published data, or is this new data?
Response >> This figure is a new data. However, sections re-analyzed for this graph were same with the previous research.
Figure 2 legend: please add a citation to the source publication. Is this data directly from Casanova-Acebes 2021, or did the authors perform new analyses of the published scRNAseq data?
Response >> Following sentence was added in the figure legend.
“Re-analysis of published single-cell RNA sequence data for NSCLC (BioProject accession; PRJNA609924) [69].”
Figure 3: Is the quantitative data in panels C and D derived from the single tissue section shown in A and B? Is this previously published or new data?
Response >> This figure is a new figure, not published in previous research. However, sections were same with the previous research.
Figure 4: What is known regarding mechanisms by which exogenous SPP1 induces chemoresistance in cancer cells? Is a specific receptor involved? Based on the cited antibody blocking, is integrin αvβ3 the relevant receptor?
Response >> Thanks for a critical point. CD44, rather than integrin αvβ3 was suggested to be involved in chemo-resistance. This point was described in ref 75.
The sentence was changed as follow.
“SPP1 facilitated cancer cell chemoresistance via the activation of the CD44 receptor, PI3K/AKT signaling, and ABC drug efflux transporter activity, and anti-SPP1 and anti-CD44 antibody improved the cancer cell sensitivity to cisplatin in mouse model [75].”
LLC is used without definition. Lewis lung carcinoma?
Response >> “Lewis lung carcinoma” was added in the text.
Line 360: is a word missing in “In breast cancer, SPP1 derived from TAMs was suggested to cancer cell growth and progression”
Response >>The sentence was changed as follow.
“In breast cancer, SPP1 derived from TAMs was suggested to be involved in cancer cell growth and progression”
Line 364 please correct “High density of SPP1-expres-364 sin TAMs and CAF”
Response >>The sentence was changed as follow.
“High densities of SPP1-expressing TAMs and CAF”
Need to correct several grammar/usage errors including in line 399
Response >> English check was re-performed by a native English speaker.
The data availability statement may need to be modified unless proper citations are provided for what appear to be primary data presented in the figures. At least need to provide GEO accession numbers for the relevant RNAseq data.
Response >>Following sentence was added in the figure legend.
“Re-analysis of published single-cell RNA sequence data for NSCLC (BioProject accession; PRJNA609924) [69].”
Reviewer 3 Report
Journal: Cancers (ISSN 2072-6694)
Manuscript Title: The significance of SPP1 in lung cancers and its impact as a marker for protumor tumor-associated macrophages
Manuscript ID: cancers-2270634
Submission Date: Friday, March 24, 2023
The manuscript presented “a study on the SPP1 significance in lung cancers”. However, the major and critical weak points are that:
(1) Their proposed work discussion is weak distributed to be described or analyzed.
(2) The novelty is not guaranteed.
(3) Their work is not compared with state-of-the-art approaches nor related studies.
(4) Their experiments leak from the descriptive and statistical analysis.
The rest of my review presents other weak points, comments, and opinions in detail.
Overall Comments:
(1) [ABSTRACT] The abstract should reflect the contributions of the manuscript. I suggest rewriting it.
(2) [INTRODUCTION] The authors should provide a clear problem definition and contributions in the introduction section.
(3) [RESEARCH QUESTION] Where is the research question and research gap?
(4) [RESEARCH QUESTION] The research question is not well-formulated or is poorly motivated, and the paper does not provide new insights or information that is not already known.
(5) [RELATED WORK] Where are the related studies? They should be declared in a separate section.
(6) [RELATED WORK] A table of comparisons should be added at the end of the related studies section to praise the pros. and cons. of them. The year column should be added and they should be ordered by it.
(7) More details and comparisons should be presented.
(8) [CONCLUSIONS] The conclusions in this manuscript are primitive. Please, write your conclusions.
(9) [REFERENCES] There are no citations for many sentences in the manuscript. Why? Please check.
(10) [REFERENCES] The references should be written in the same style following the journal authors’ guidance.
(11) [REFERENCES] Recent citations from 2021 to 2023 should be added to the manuscript.
(12) [PROOFING] The authors should get editing help from someone with full professional proficiency in English.
(13) [PROOFING] The manuscript should be checked again to fix any typos such as missing spaces and commas.
(14) [CONSISTENCY] The manuscript structure is too short. It must be elaborated in their applied technology as should support more rigorous technical aspects.
(15) [CONSISTENCY] Some paragraphs are wrapped in more than 10 lines. They should be split concisely.
(16) [NOVELTY] What is the novelty of the suggested approach?
(17) [FIGURES] The authors should provide high-resolution figures in the manuscript. For example, Figure 3.
(18) [LIMITATIONS] What are the limitations of the current study? It should be added in a separate section.
For the authors in case of the authors got a chance to review the manuscript and submit the revised one after the editor’s decision, please, provide a table in the revised manuscript mentioning (1) the comment, (2) the authors’ response, and (3) the authors’ change (if applicable). Please, consider all of the comments and don’t ignore any of them.
Please, refer to the attached file "cancers-2270634 Reviewer.pdf" for the same comments in an organized format.

Author Response
Manuscript Title: The significance of SPP1 in lung cancers and its impact as a marker for protumor tumor-associated macrophages
Manuscript ID: cancers-2270634
Submission Date: Friday, March 24, 2023
The manuscript presented “a study on the SPP1 significance in lung cancers”. However, the major and critical weak points are that:
(1) Their proposed work discussion is weak distributed to be described or analyzed.
(2) The novelty is not guaranteed.
(3) Their work is not compared with state-of-the-art approaches nor related studies.
(4) Their experiments leak from the descriptive and statistical analysis.
The rest of my review presents other weak points, comments, and opinions in detail.
Overall Comments:
(1)[ABSTRACT] The abstract should reflect the contributions of the manuscript. I suggest rewriting it.
Response >> We sincerely appreciate for kind and critical comments. Some words were added in the abstract.
(2)[INTRODUCTION] The authors should provide a clear problem definition and contributions in the introduction section.
Response >> We sincerely appreciate for kind and critical comments. Some words and sentences were added in the introduction.
(3) [RESEARCH QUESTION] Where is the research question and research gap?
(4) [RESEARCH QUESTION] The research question is not well-formulated or is poorly motivated, and the paper does not provide new insights or information that is not already known.
Response >>Following sentences were added in the introduction.
“In this review, we summarize the significance of TAMs in lung cancers and discuss the importance of SPP1 as a new marker for the protumor subpopulation of TAMs in lung adenocarcinoma. In addition, SPP1 has been suggested to be useful in distinguishing monocyte-derived TAMs from resident TAMs. The significance of granulocyte macrophage colony stimulating factor (GM-CSF)-related signals and SPP1/CD44 axis in chemoresistance is also discussed.”
(5) [RELATED WORK] Where are the related studies? They should be declared in a separate section.
Response >>Most critical related work is ref 17; Cancers 2022, 14(18), 4374. doi: 10.3390/cancers14184374.
As reviewer commented, we thought to divide this content to a separated section. However, this article is composed of in vivo study and in vitro studies using many cell types, and it was difficult to separate the section. No comments on this point were raised by other two reviewers.
(6)[RELATED WORK] A table of comparisons should be added at the end of the related studies section to praise the pros. and cons. of them. The year column should be added and they should be ordered by it.
Response >> Thanks for your important comment. New table was added in the file.
(7) More details and comparisons should be presented.
Response >> Additional table provided a brief comparisons of previous works.
(8) [CONCLUSIONS] The conclusions in this manuscript are primitive. Please, write your conclusions.
Response >>In conclusion (section 9), sentences was changed as follows;
“In this review, we discussed the significance of TAM-derived SPP1 in the progression of lung adenocarcinoma. SPP1 is not only a prognostic marker for lung adenocarcinoma, but also a marker for monocyte-derived macrophages in lung adenocarcinoma. As SPP1 expression in TAMs is induced by GM-CSF, and GM-CSF is produced by cancer cells stimulated with anti-cancer agents, cell-to-cell communication via GM-CSF and SPP1 is thought to be involved in the development of chemoresistance. SPP1 is a critical factor in the development of a protumor TME. Thus, targeting SPP1-related signaling pathways such as SPP1/CD44 axis is a promising approach for the treatment of lung adenocarcinoma.”
(9) [REFERENCES] There are no citations for many sentences in the manuscript. Why? Please check.
Response >>When two or three sentences were contiguous, we added the reference number only in the first sentences. But we added the reference number in all sentences as you recommended.
(10) [REFERENCES] The references should be written in the same style following the journal authors’ guidance. (11) [REFERENCES] Recent citations from 2021 to 2023 should be added to the manuscript.
Response >>These points were corrected in the revised version.
(12) [PROOFING] The authors should get editing help from someone with full professional proficiency in English. (13) [PROOFING] The manuscript should be checked again to fix any typos such as missing spaces and commas.
Response >> English check was re-performed by a native English speaker.
(14) [CONSISTENCY] The manuscript structure is too short. It must be elaborated in their applied technology as should support more rigorous technical aspects.
Response >> The words of this manuscript is more than 4000 words, and we are now thinking that this review article is somewhat long.
(15) [CONSISTENCY] Some paragraphs are wrapped in more than 10 lines. They should be split concisely.
Response >> We separated some paragraph, as reviewer commented. Paragraph was changes 9 to 12.
(16) [NOVELTY] What is the novelty of the suggested approach?
Response >> As reviewer 2 also similar comments, so we added some sentences in the text.
(17) [FIGURES] The authors should provide high-resolution figures in the manuscript. For example, Figure 3.
Response >> We changed the figure to new one. But please not picture figures often change in review version from original submitted version.
(18) [LIMITATIONS] What are the limitations of the current study? It should be added in a separate section.
Response >> Following paragraph was added.
“11. Limitations
In this review, we mainly discussed the function and significance of SPP1 derived from TAMs in lung cancer, but differences might exist between each organ and histological subtype. In addition to this discrepancy in organs, a discrepancy between humans and mice is also indicated.”
Round 2
Reviewer 3 Report
Journal: Cancers (ISSN 2072-6694)
Manuscript Title: The significance of SPP1 in lung cancers and its impact as a marker for protumor tumor-associated macrophages
Manuscript ID: cancers-2270634 R1
Submission Date: Monday, April 3, 2023
The authos made a suitable revision. However, there are some comments that need to be addressed:
Overall Comments:
(1) Table 1: give the authors credits by mentioning their names in the first column beside their citations. For example, xxx et al. [xx].
(2) Where are the figures in the updated version?
(3) The * in the affiliation need to be checked again because they are not the “first” two authors. I think it should be “Both authors contributed equally to this work.”.
(4) What is the meaning of “pStage” in the third column in Table 1?
For the authors in case of the authors got a chance to review the manuscript and submit the revised one after the editor’s decision, please, provide a table in the revised manuscript mentioning (1) the comment, (2) the authors’ response, and (3) the authors’ change (if applicable). Please, consider all of the comments and don’t ignore any of them.
Please, refer to the attached file "cancers-2270634 R1 Reviewer.pdf" for the same comments in an organized format.

Author Response
Overall Comments:
(1) Table 1: give the authors credits by mentioning their names in the first column beside their citations. For example, xxx et al. [xx].
Response>> We sincerely appreciate for kind reviewing, Table 1 was corrected as reviewer commented.
(2) Where are the figures in the updated version?
Response>> Corrected version of figure 1D was up-loaded.
(3) The * in the affiliation need to be checked again because they are not the “first” two authors. I think it should be “Both authors contributed equally to this work.”.
Response>> This point was corrected.
(4) What is the meaning of “pStage” in the third column in Table 1?
Response>> “pStage” was corrected as “Stage”.